# Improvement of Temperature Control Performance of Thermoelectric Dehumidifier Used Industry 4.0 by the SF-PI Controller

**Jae-Sub Ko [1], Jun-Ho Huh [2] and Jong-Chan Kim [3,*]**

[1]    Department of Electric Control Engineering, Sunchon National University, 255 Jungang-ro, Suncheon-city, Jeollanam do 57922, Korea; kokos22@sunchon.ac.kr

[2]    Department of Software, Catholic University of Pusan, Geumjeong-gu, 57 Oryundae-ro, Geumjeong-gu, Busan 46252, Korea; 72networks@pukyong.ac.kr or 72networks@cup.ac.kr

[3]    Department of Computer Engineering, Sunchon National University, 255 Jungang-ro, Suncheon-city, Jeollanam do 57922, Korea

*    Correspondence: seaghost@sunchon.ac.kr; Tel.: +82-61-750-3620

**Abstract:** This paper proposes the series connected fuzzy-proportional integral (SF-PI) controller, which is composed of the fuzzy control and the PI controller to improve temperature control performance of dehumidifier using a thermoelectric element. The control of conventional PI controller usually uses fixed gain. For that reason, it is limited in achieving satisfactory control performance in both transient-state and steady-state. The fuzzy control within SF-PI controller adjusts the input value of PI controller according to operating condition. The PI controller within the SF-PI controller controls the temperature of the thermoelectric element using that value. The SF-PI controller can achieve more accurate temperature control than a conventional PI controller for that reason. The SF-PI controller has been tested for various indoor environmental conditions such as temperature and relative humidity conditions. The average temperature error of the SF-PI controller between the reference temperature and the thermoelectric element temperature is 22% of traditional PI's value and consumption power is reduced by about 10%. Therefore, the SF-PI controller proposed in this paper can improved the performance of temperature control of dehumidifier using thermoelectric element. The power consumed by buildings accounts for a significant portion of the total power consumption, of which the air conditioner represents the largest energy consumer. In this paper, it is possible to reduce the energy consumption by improving the performance of the dehumidifier, one of the air conditioners, and it can be applied to various control fields in the future.

**Keywords:** thermoelectric element; fuzzy; PI controller; dehumidifier; Industry 4.0; temperature-control; smart gird; water grid; computer architecture

---

## 1. Introduction

Energy consumption is increasing globally and energy demand is expected to increase by 30% by 2040. A climate change agreement was signed in Paris in 2015 to cope with the rapidly changing climate [1,2]. The energy consumed by buildings is about 80%–85% in high temperature and high humidity regions, and 39% and 40% in Europe and the USA, respectively [3,4]. The $CO_2$ emissions from these buildings account for 30%–40% of all industries [5]. In particular, the air conditioning system occupies 10% of the total energy and in Japan, the United States, and Korea, 91%, 90%, and 86% of residents have air conditioners [6]. Such an air conditioning system is used for heating, ventilating, and cooling the room air, and it represents a financial burden for consumers since it is continuously used even in the peak load time of the building. The load on these air conditioning systems accounts

for about 25% of the building load according to the data released by the US Energy Information Administration in 2017 [2].

The dehumidifier system includes a system using a moisture absorbent and a cooling system using a compressor. Among them, the dehumidifier system using compressors is the most widely used dehumidifier system [7]. This method has a similar structure and principle to the air conditioner [8]. Therefore, the power consumption of the dehumidifier is almost the same as that of the air conditioner. The difference between a dehumidifier and an air conditioner is the configuration of the condenser and vaporizer. The dehumidifier is composed of one unit, but the air conditioner is installed separately. The major objective of the dehumidifier is to remove moisture from the air, so the air passing through the vaporizer is slower than the air conditioner. Advantages of the compressor dehumidifier are that they are excellent in dehumidification performance in a climate of 15 °C or higher and can be maintained only by increasing the room temperature by 1–2 °C without cooling indoor air like an air conditioner. However, when the temperature of the air for dehumidification is 15 °C or less, the temperature of the vaporizer is close to the freezing point, and ice is formed on the coil of the vaporizer, thereby deteriorating the dehumidification performance. It also consumes 67% of time and energy to defrost the coil. In addition, the compressor cooling method is generally very heavy (weight of 10 kg or more), large in terms of volume, and noisy (more than 40 decibels) [7]. Therefore, the energy saving of an air conditioning system is essential to reducing the greenhouse gas emissions generated by the power consumption in buildings.

The Fourth Industrial Revolution is characterized by technology convergence, and the Fourth Industrial Revolution includes technologies such as Big Data Analysis, Artificial Intelligence, Robotics, and the internet of thing (IoT). Among them, the IoT is a function that can transmit data by connecting everything through a network regardless of the kind of thing. The IoT system can be applied to various fields by constituting the sensor, the actuator, and the control units. IoT systems can take advantage of the embedded hardware platform, and their interest in the open source platform is increasing due to their low cost, computational power, programming flexibility, I/O interface, low energy consumption, and reduced size [9,10]. Among the open source platforms, the Raspberry Pi and the Arduino are the most favored and widely studied and applied platforms for Industry 4.0 applications [9–14]. Recently, as the importance of indoor environment increases, interest in systems that can automatically control or check the indoor air state is increasing. Therefore, Industry 4.0 is widely applied to home appliances. To control systems automatically, it has to use a control method [15,16].

The methods used in automatic control systems use the Proportional Integral Derivative (PID) [17,18], Neural-Network [19,20], Fuzzy control [21,22], Genetic algorithm [23,24], etc. Among them, PID control is most widely used in the industrial field [25–27]. PID control has various advantages such as a simple structure, an excellent control performance, and easy adjustment of control gain. Furthermore, PI control is excellent in reducing errors in steady-state [28]. The PI controller predicts the control value of the next step using a proportional gain and an integral gain. The PI controller has good control performance for particular gains, however, if these gains are not selected an optimization value suitable for control condition, the performance of control will become worse. Therefore, it is necessary to study the method for reducing the dependence of gain of PI controller [29].

In this paper, the Series Connected Fuzzy-PI (SF-PI) controller controls temperature of a dehumidifier using the thermoelectric element. The conventional PI controller has a limitation in improvement of the control performance in transient-state due to the fixed gain value. To solve this problem, the SF-PI controller is configured, connecting the fuzzy controller and PI controller in series. The Fuzzy controller adjusts the input value of the PI controller, and then the PI controller controls the temperature of the thermoelectric element using the input value adjusted by the fuzzy controller.

This paper is organized as follows. In Section 2, the characteristics of thermoelectric devices are introduced. Section 3 shows the SF-PI (Series Fuzzy-PI) controller in which fuzzy control and PI controller are connected in series. In Section 4, Industry 4.0 implementation using web page

and temperature control experiment result by SF-PI controller are presented and the validity of the proposed method is analyzed. Finally, Section 5 presents the conclusion and future work of this paper.

## 2. Dehumidifier System by Thermoelectric Element

Humid air entering the interior by the fan contacts the surface of the cooler and condenses into water [30]. In general, a dehumidifier removes moisture in the air by this method and discharges the dehumidified air. Dried exhaust air is used to cool the temperature of the refrigerant used to drop the temperature of the cooler. As a result, the temperature of the output air is raised. Figure 1 shows the principle of the dehumidifier. Figure 2 also shows the structure of the thermoelectric element.

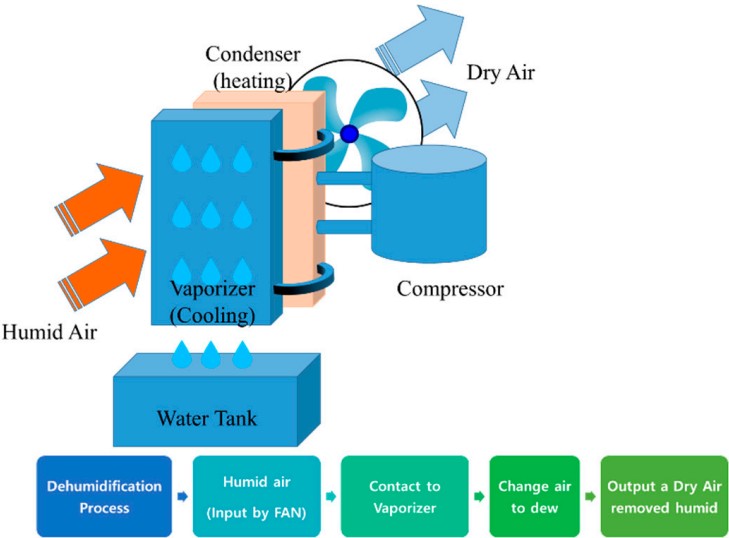

**Figure 1.** Principle of the dehumidifier.

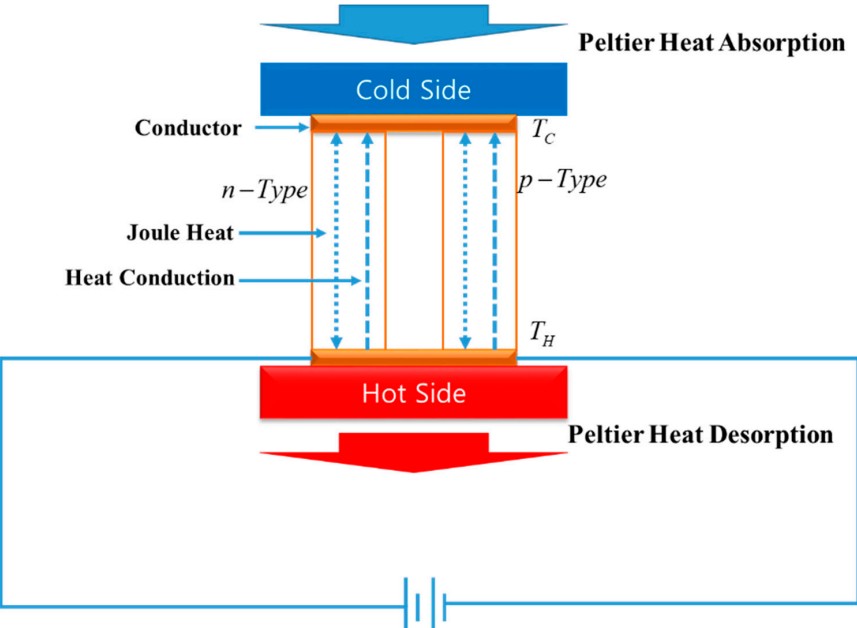

**Figure 2.** Principle of thermoelectric element.

When a current flows through a thermoelectric element, one side of the thermoelectric element made of P-N semiconductor absorbs heat and the other side outputs heat. The Joule's heat caused by the current (I) flowing into the thermoelectric element each enters the top and bottom side. Therefore,

the top side has low temperature ($T_c$), and bottom side has high temperature ($T_H$). The temperature differential is as follows:

$$\Delta T = T_H - T_c \tag{1}$$

The amount of heat absorption ($Q_c$) on the low temperature side and radiation of heat ($Q_H$) on the high temperature side are as follows [31]:

$$Q_c = \propto T_c I - \frac{I^2 R}{2} - K\Delta T \tag{2}$$

$$Q_H = \propto T_H I + \frac{I^2 R}{2} - K\Delta T \tag{3}$$

where $\alpha$ (V/K) is the Seebeck coefficient, $R$ ($\Omega$) is the inner resistor of thermoelectric element, and $K$ (W/m·K) is the coefficient of thermal conductivity.

The first term is the influence of the Peltier effect, the second term is the influence of the Joule's effect and the last term is the amount of thermal transfer by conduction from the high temperature side to the low temperature side in Equations (2)–(3).

The input power of the thermoelectric element is as follows:

$$P_{in} = \propto I(T_c - T_H) + I^2 R \tag{4}$$

The first term is the power by the Seebeck effect, and the second term is the power by the electric resistor in Equation (3). The sum of these powers becomes the total input power of the thermoelectric element.

The coefficient of performance (*COP*) of the thermoelectric element is the ratio of heat absorption to the input power, and is as follows [32]:

$$COP = \frac{Q_c}{P_{in}} \tag{5}$$

## 3. SF-PI Control System

The PI controller is widely used in industrial fields because it has advantages in that the structure is simple and the response between the control and the control value is clear.

The typical PI controller calculates the control value using an error signal, a proportional ($k_p$) and integral gain ($k_i$) or integral time ($T_i$). The optimal gains of PI controller change by the operating state. Therefore, gain adjustment is necessary, and the adjustment has to consider the control characteristics suitable to the system. The characteristics requested to the PI controller are as follows.

- Stable performance
- Fast response
- Minimize errors in the steady-state

The $k_p$ and $k_i$ ($T_i$) must be adjusted within a stability performance range. In generally, the increasing of gain $k_p$ and $k_i$ (decreasing of the integral time constant $T_i$) brings the system the quick response. However, excessive increase or decrease of these values causes vibration of the system. Therefore, the system becomes a divergent system at worst.

There is a close relationship between system performance and these gains, so the gains have to be adjusted suitably. These gains generally have the following influence on the system (Table 1).

**Table 1.** The effect of the proportional integral (PI) controller gain value on the system.

| Parameter | Rise Time | Overshoot | Settling Time | Steady-State Error |
|-----------|-----------|-----------|---------------|--------------------|
| $k_p$ Increase | Decrease | Increase | Small Change | Decrease |
| $k_i$ Increase ($T_i$ Decrease) | Decrease | Increase | Increase | Decrease Significantly |

Table 2 shows the various ways to adjust the gain of the PI controller [33,34]. This paper adjusts the gain of the PI controller through the trial and error method, which is one of the manual methods that does not need mathematical expression and can control the gain value online.

**Table 2.** Methods of adjusting the gain of a PI controller.

| Method | Advantages | Disadvantages |
|--------|-----------|---------------|
| Manual | Online method<br>No math expression | Requires experienced personnel |
| Ziegler-Nichols | Online method<br>Proven method | Some trial and error, process upset<br>and very aggressive tuning |
| Cohen-Coon | Good process models | Offline method<br>Some math<br>Good only for first order processes |
| Software tools | Online or Offline method, consistent tuning,<br>supports Non-Steady State tuning | Some cost and training involved |
| Algorithmic | Online or offline method, consistent tuning,<br>supports Non-Steady State tuning, very precise | Very slow |

The adjusting method of $k_p$ and $k_i$ ($T_i$) is as follows.

First, after the command value is changed, if the response time is very slow, increase $k_p$, and if the response time is fast but unstable, decrease $k_p$.

Second, if the feedback value does not track the command value, decrease $T_i$, and if the feedback value unstably tracks the command value with vibration, increase $T_i$.

The PI controller used in this paper adjusts $k_p$ and $k_i$ using the manual method.

The manual method adjusts the PI controller with the procedure shown below.

1. Proportional gain setting.

The control is started with $k_p = 0$, without the $k_i$ gain. Increase $k_p$ until the control loop achieves good (satisfactory) stability.

2. Integral time ($T_i$) & $k_p$ adjustment.

Set the integral time $T_i$, which is the time between the over-shoot and the under-shoot of the step response, with the proportional controller. Decrease $T_i$ until the rising time achieves a satisfactory state. If the control state is unstable during the $T_i$ setting, decrease $k_p$.

The PI controller consists of a proportional control and an integral control. Among them, the integral operation causes the cumulative error due to the steady state. This problem causes saturation problems in the system. Therefore, the PI controller results in a control delay when the system is reversed until the cumulative error is removed. This phenomenon is called the windup. The avoiding method of the windup is called the anti-windup [35–37].

In this paper, anti-windup is applied using output limit. The problem of windup is that the output of the accumulator integrator reduces the influence of the proportional controller in the next state, resulting in delayed control. Therefore, the method used in this paper limits the output value from the PI controller and continues to use the limited value when calculating the next state value. Therefore, even if the state changes to the next state, the influence of the proportional controller output can be maintained because the output of the PI controller has a limited value. The following figure shows the

PI controller used in this paper. The PI controller outputs the change value of the control signal with the error compensated by the fuzzy controller as an input, and outputs the control value of the next state in addition to the output value of the previous state. At this time, since the control value of the next calculated state is limited, the PI controller can maintain the influence of the proportional control. Figure 3 shows a PI controller with the anti-windup.

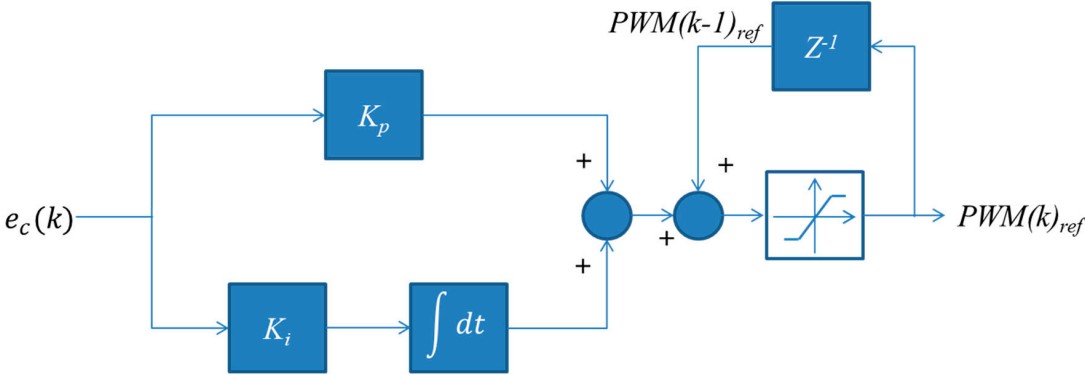

**Figure 3.** PI controller with anti-windup.

The purpose of this paper is to improve the temperature control performance of a dehumidifying system using a thermoelectric device. In this paper, we design a system to control the input value of the PI controller because the PI controller has a limitation in the performance improvement due to the fixed gain value. Control of input values of PI controller uses fuzzy control that does not require mathematical modeling and robustness to a nonlinear system.

The PI controller controls the pulse width modulation (PWM) for the thermoelectric device control, and the fuzzy control controls the input value of the PI controller. In this paper, fuzzy control and PI controller are arranged and connected in series to construct a system that operates like this.

The fuzzy control is performed by inputting the error ($e$) and the changing error ($ce$) values by the reference temperature ($T_{ref}$) and the heat absorption side temperature of the thermoelectric element. If the dew point temperature ($T_{DP}$) is used as the reference temperature for the thermoelectric device heat absorption side (HAS) temperature control, the change in temperature and relative humidity due to the ambient environment will greatly affect the dehumidification performance. Therefore, the reference temperature is set using the band gap temperature ($\Delta T$), and the HAS temperature is controlled below that temperature.

If two or more thermoelectric elements are used in the dehumidifier, the temperature of each HAS is different. The HAS temperature is most closely related to the dehumidification performance, and if the HAS temperature is higher than the dew point temperature, dehumidification is not performed.

Therefore, each HAS temperature is measured and the temperature control is performed based on the higher HAS temperature. A block diagram of this system is shown in Figure 4.

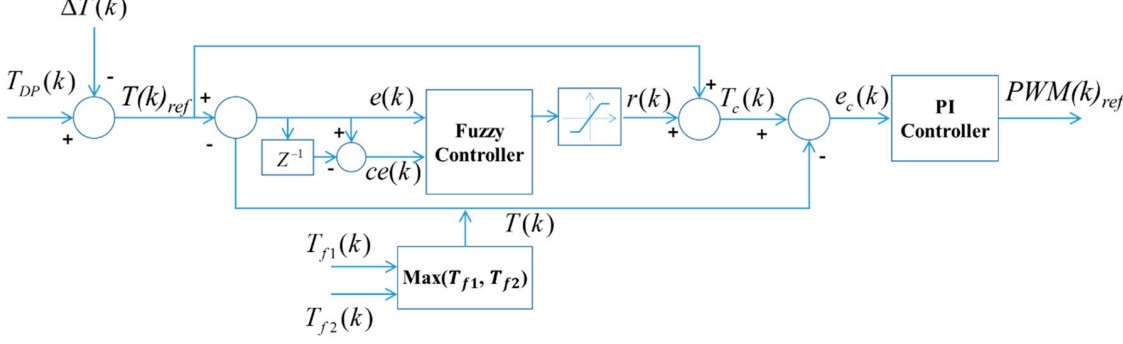

**Figure 4.** Series connected Fuzzy (SF)-PI controller.

$T_{DP}$ is the dew point temperature according to humidity and current temperature. $\Delta T$ is the setting temperature to control the temperature below the dew point temperature. $T_{ref}$ is the reference temperature for dew point temperature control. $T_{f1}$ and $T_{f2}$ are the temperature of heat absorption side in the thermoelectric element. If the thermoelectric element is over two, the input value of the controller is the temperature of the heat absorption side of the thermoelectric element that has high temperature.

$$T(k)_{ref} = T_{DP}(k) - \Delta T(k) \tag{6}$$

$$T(k) = \max\left(T_{f1}, T_{f2}\right) \tag{7}$$

$$e(k) = T(k)_{ref} - T(k) \tag{8}$$

$$ce(k) = e(k) - e(k-1) \tag{9}$$

The input value of fuzzy control is an error ($e$) and a changing error ($ce$), and its output is a value ($rc$) to adjust the input value of PI controller. The fuzzy rule base is as Table 3, and Figures 5 and 6 show the membership function for input and output value of fuzzy control.

**Table 3.** Rule base of Fuzzy controller.

| $ce$ \ $e$ | NB | NM | NS | Z | PS | PM | PB |
|---|---|---|---|---|---|---|---|
| NB | NB | NB | NB | NB | NM | NS | Z |
| NM | NB | NB | NB | NM | NS | Z | PS |
| NS | NB | NB | NM | NS | Z | PS | PM |
| Z | NB | NM | NS | Z | PS | PM | PB |
| PS | NM | NS | Z | PS | PM | PB | PB |
| PM | NS | Z | PS | PM | PB | PB | PB |
| PB | Z | PS | PM | PB | PB | PB | PB |

The membership functions in fuzzy control are formed using straight or curved lines. The triangular membership function and trapezoidal membership function in straight membership function are the most common methods. Due to their simple formulas and computational efficiency, these methods have been used widely. The performances of the membership functions have compared about vary control and the triangular and trapezoidal types can obtain benefits to real time control. The advantages of the triangular and trapezoidal types are proposed by many studies [38–42]. Furthermore, since the triangular type has robustness in terms of the response speed and the error in steady-state, it has used extensively in the related region [38,39]. Therefore, the fuzzy control used in this paper has applied the triangular membership function.

The $rc$ value is obtained in Equation (10) using the center-of-gravity defuzzification method.

$$rc(k) = \frac{\sum_{i=1}^{n} (rc)_i \cdot \mu[(rc)_i]}{\sum_{i=1}^{n} \mu[(rc)_i]} \tag{10}$$

$$T_c(k) = T(k)_{ref} + rc(k) \tag{11}$$

$$e_c(k) = T_c(k) + T(k) \tag{12}$$

$$PWM(k)_{ref} = PWM(k-1)_{ref} + k_p \cdot \{e_c(k) - e_c(k-1)\} + k_i \cdot e_c(k) \cdot t_s \tag{13}$$

$T_c$ is the reference value adjusted by the fuzzy control, $e_c$ is the input value of PI controller, and $k_p$ and $k_i$ are the proportional gain and integral gain used in PI controller, respectively.

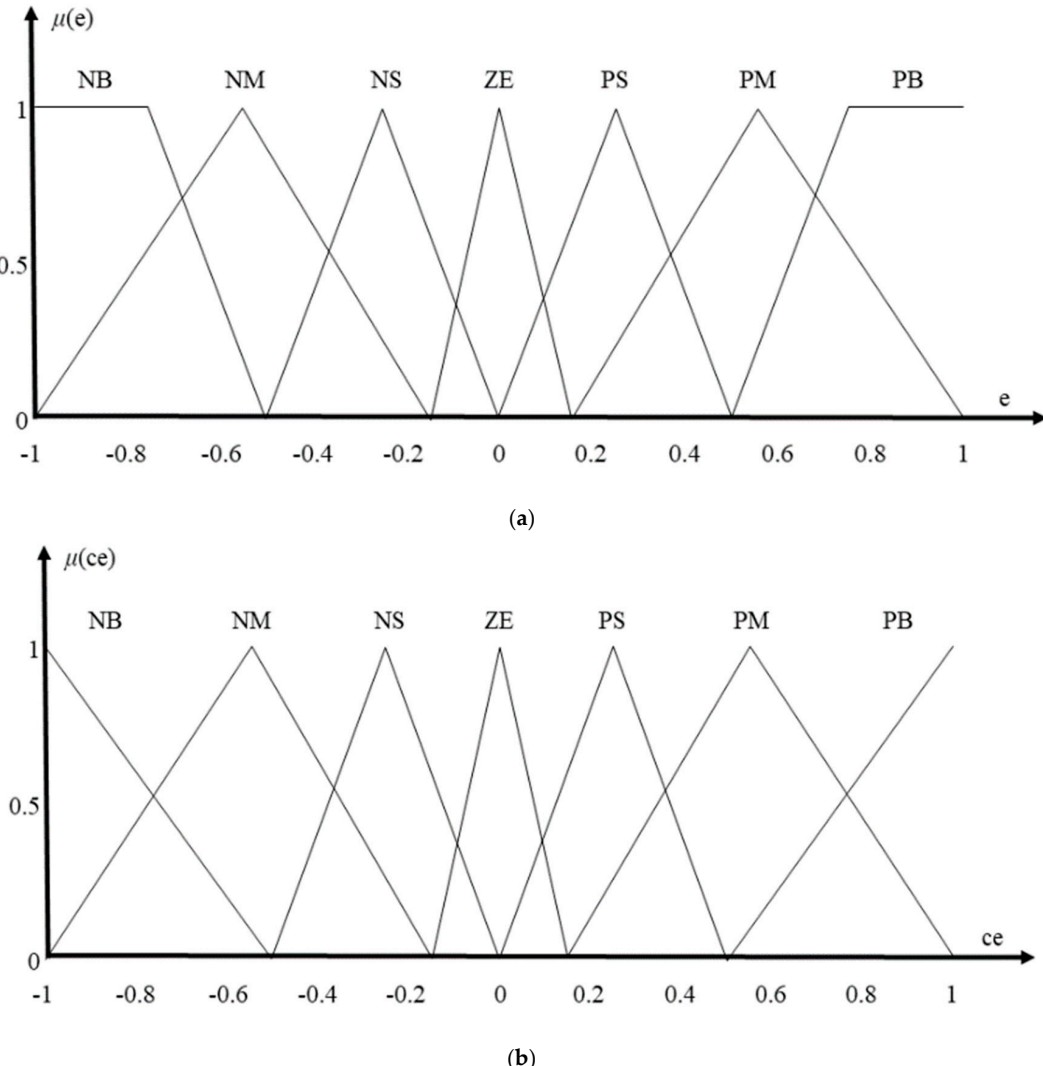

(**a**)

(**b**)

**Figure 5.** Membership function for the input value (error (*e*) and changing error (*ce*)). (**a**) The error (*e*) membership function. (**b**) The changing error (*ce*) membership function.

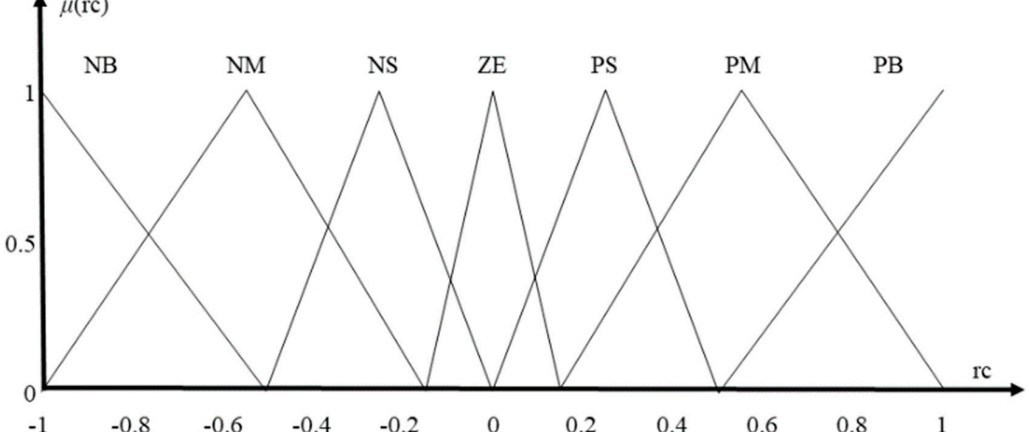

**Figure 6.** Output membership function. NB: Negative Big, NM: Negative Medium, NS: Negative Small, ZE: Zero; PB: Positive Big, PM: Positive Medium, PS: Positive Small.

## 4. Experiment Result

Raspberry pi and the Arduino were used in this paper for the implementation of the IoT, one element of Industry 4.0. Raspberry pi builds the Apache HTTP server (Version: 2.2.22, Apache software Foundation, ASF, Forest Hill, MD, USA) and configures the database with MySQL (Version: 5.6.23, MySQL AB, Cupertino, CA, USA) to save the data needed for control. Arduino measures the temperature inside the dehumidifier, stores it in the database (DB), and controls the thermoelectric elements inside the dehumidifier.

Figure 7 shows a web page for the remote control. In addition, Table 4 shows the explanation of remote web page.

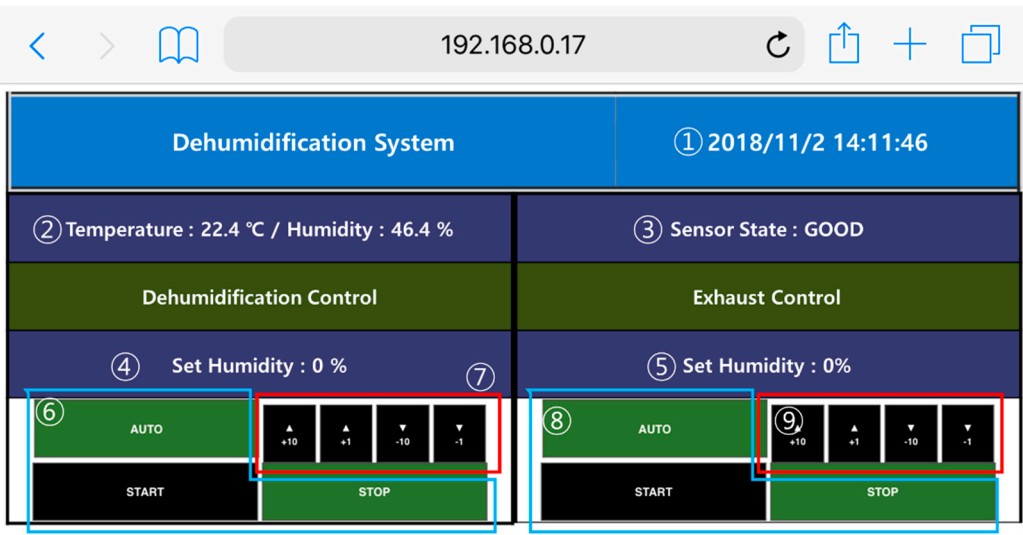

**Figure 7.** Web page for remote control.

**Table 4.** Operating explanation on web page.

| No. | Operation | Operating Explanation |
|---|---|---|
| ① | Current Time | Current time of dehumidifier |
| ② | Temperature, humidity | Temperature and humidity of intake air |
| ③ | Sensor State | Operating state of a temperature and humidity sensor<br>- Sensor normal operation: GOOD<br>- Sensor fault: ERROR (last operating time)<br>ex) ERROR(2018/11/2/ 12:11:11) |
| ④ | Dehumidification control set humidity | Humidity setting for auto dehumidification control |
| ⑤ | Exhaust control set humidity | Humidity setting for auto exhaust control |
| ⑥⑧ | Dehumidification/ Exhaust control mode setting | Mode selection for dehumidification/Exhaust<br>- AUTO: automatically control using set humidity, current humidity and temperature<br>- START: forced starting<br>- STOP: forced stopping |
| ⑦⑨ | Adjusting set humidity value | Adjusting set humidity value |

Figures 8–10, respectively show the changing of the dehumidifier operating state according to set a web page for remote control.

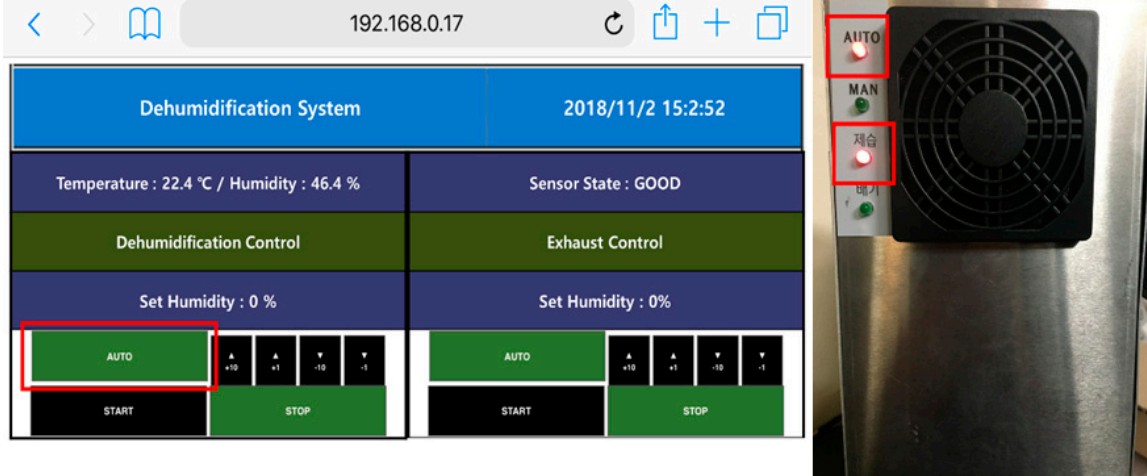

**Figure 8.** Dehumidification control—"AUTO" mode.

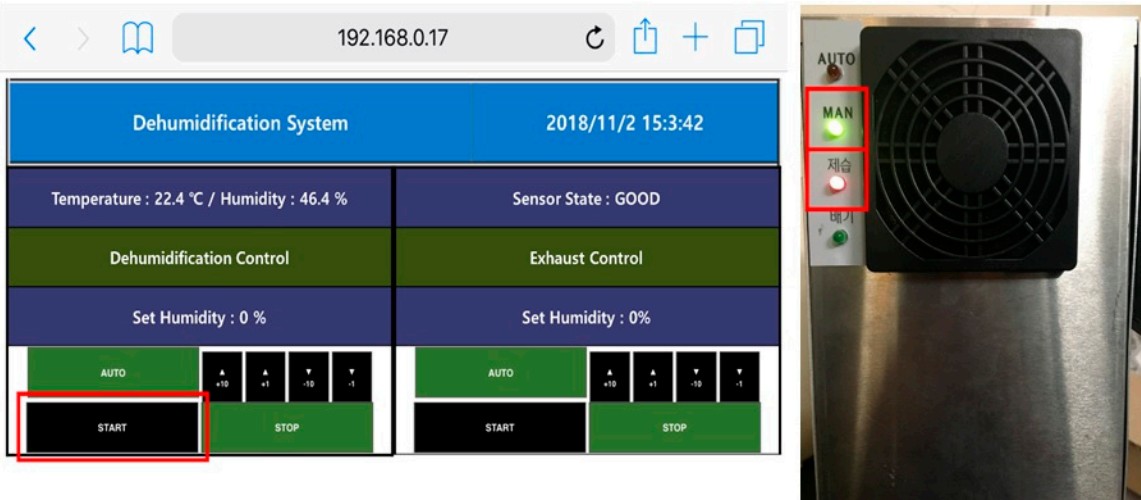

**Figure 9.** Dehumidification control—"START" mode.

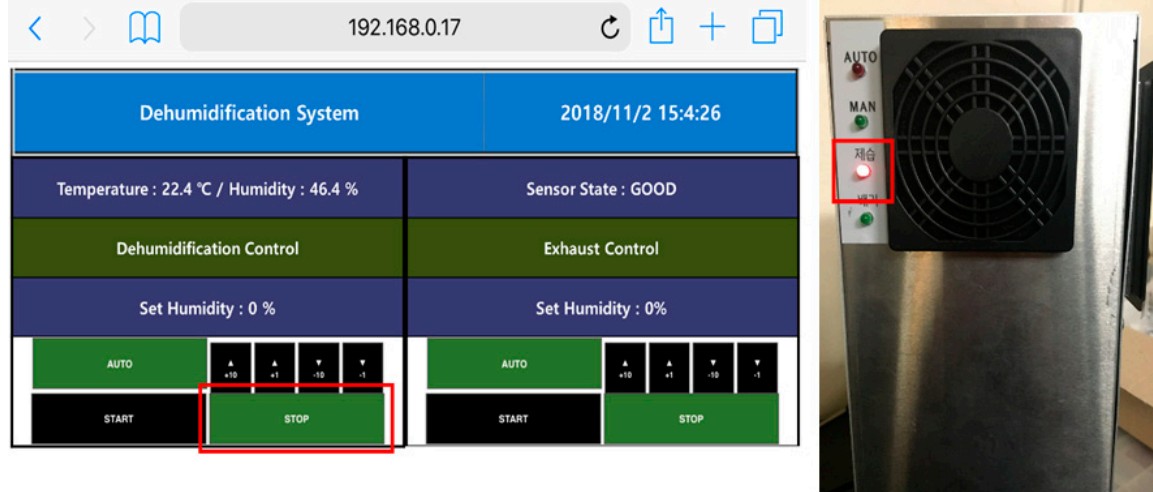

**Figure 10.** Dehumidification control—"STOP" mode.

Figure 11 shows a system structure and a circuit diagram used in this paper for experiment. Table 5 shows the major parts used in Figure 11.

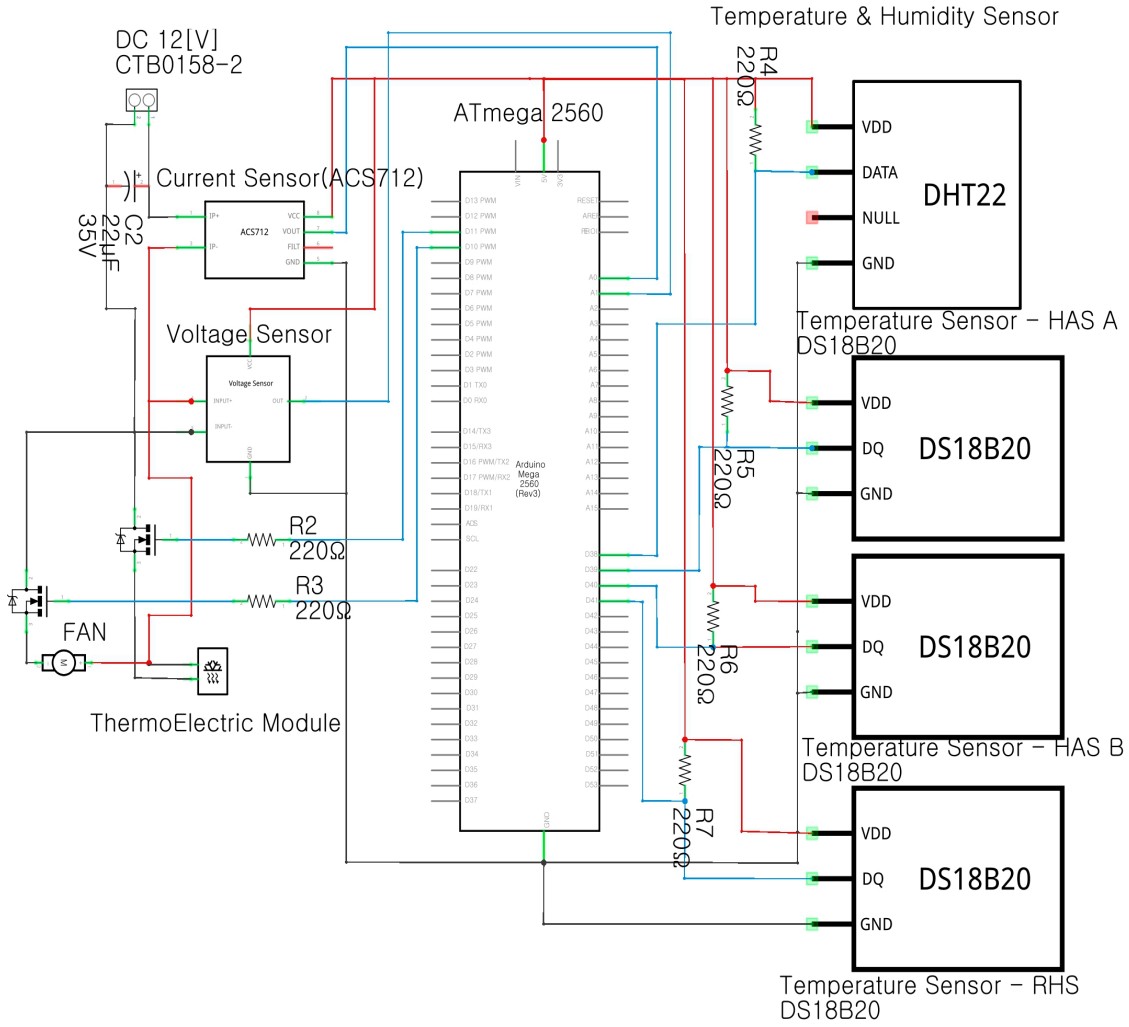

**Figure 11.** System structure and circuit diagram for experiment.

**Table 5.** Components used in the experiment.

| Component | Spec |
|---|---|
| Thermoelectric element | Model: TEC1-12708, $Q_{max}$: 71 W <br> $\Delta T_{max}$: 66 °C, $I_{max}$: 8.5 A <br> $V_{max}$: 15.4 V |
| Controller | Model: Arduino mega2560 <br> Microcontroller: ATmega2560 <br> Flash Memory: 256 KB <br> Digital I/O: 54. Analog I/O: 16 |
| Temperature & Humidity Sensor | Model: DHT22 <br> Temperature range: −40 °C–80 °C <br> Humidity range: 0–100% <br> Accuracy: Temperature (±0.5 °C), Humidity (2–5%) |
| Temperature Sensor | Model: DS18B20 <br> Temperature range: −10 °C–85 °C <br> Accuracy: ±5 °C |

In this paper, the atmospheric conditions are set as shown in Table 6 to analyze the temperature control performance of a dehumidifier using a thermoelectric device. Figures 12 and 13 show the voltage, current, and power of a typical PI controller and SF-PI. The operation of the thermoelectric device causes the current to change constantly, which also changes the power consumption. Table 7

shows the average power consumption in Figures 12 and 13. The proposed SF-PI method reduced power consumption to 93% of PI controller power consumption.

**Table 6.** Condition to analyze the control characteristics.

|  | Temperature (°C) | Relative Humidity (%) | Dew Point Temperature (°C) |
|---|---|---|---|
| Set value | 25 | 60 | 16.69 |

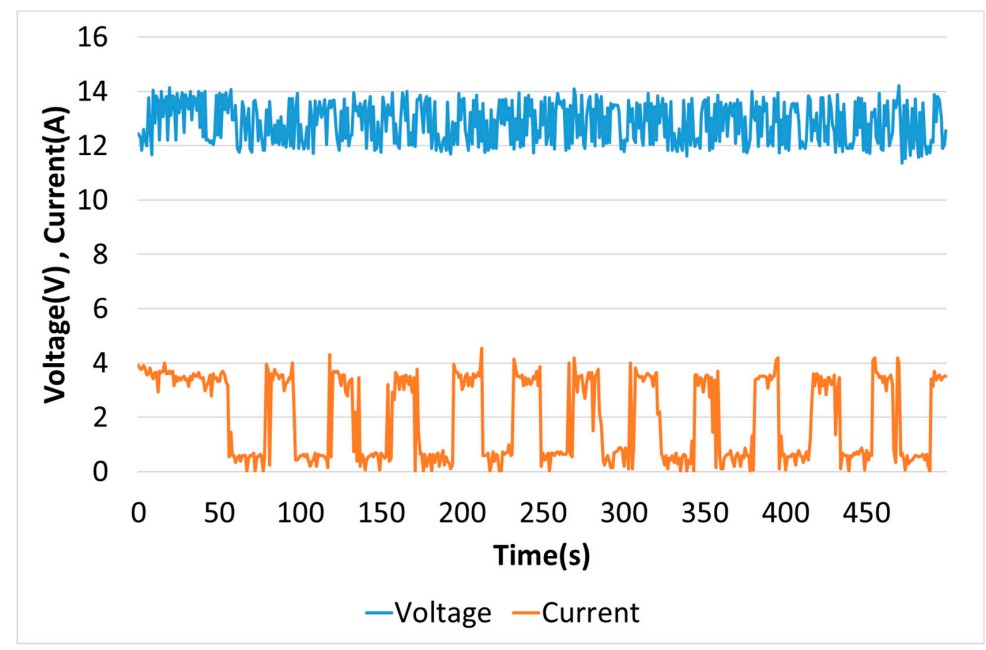

(a)

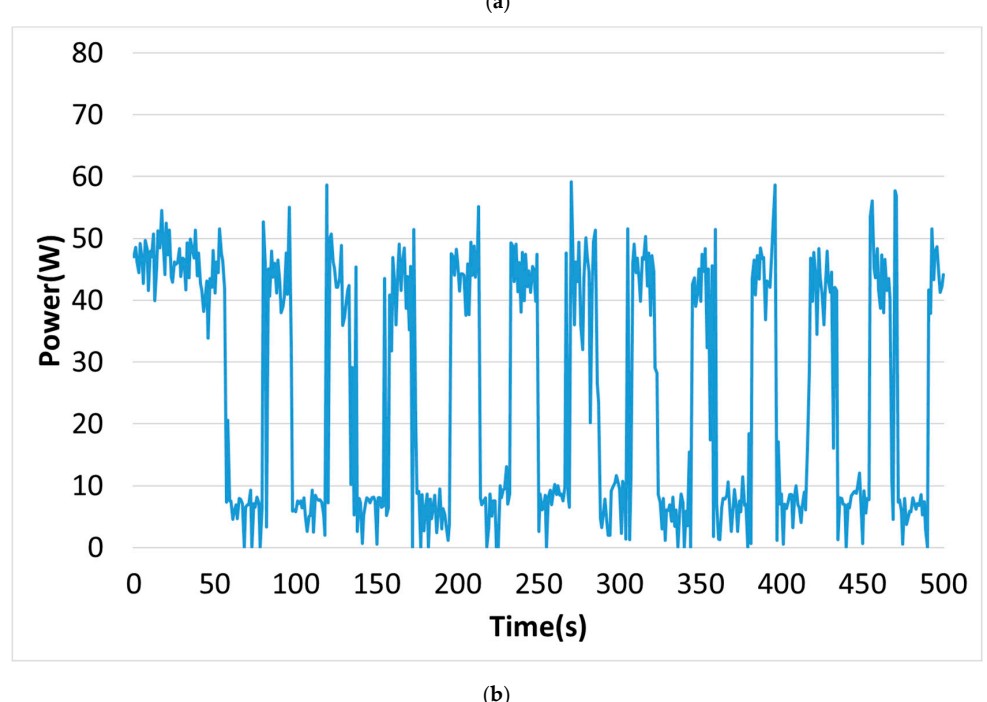

(b)

**Figure 12.** Voltage, Current, and Power of PI controller. (**a**) Voltage and Current. (**b**) Power.

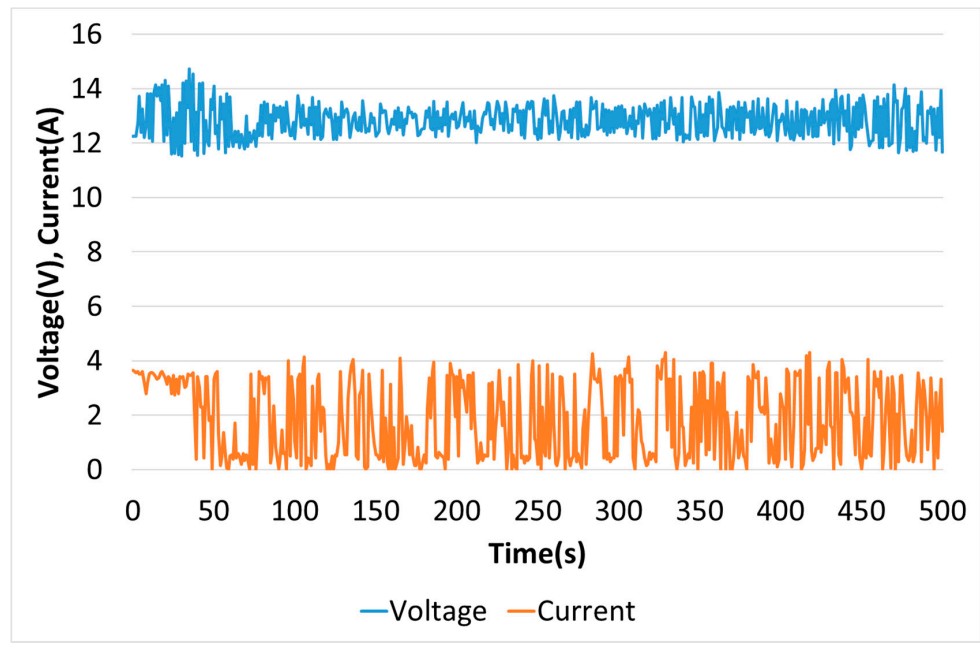

(a)

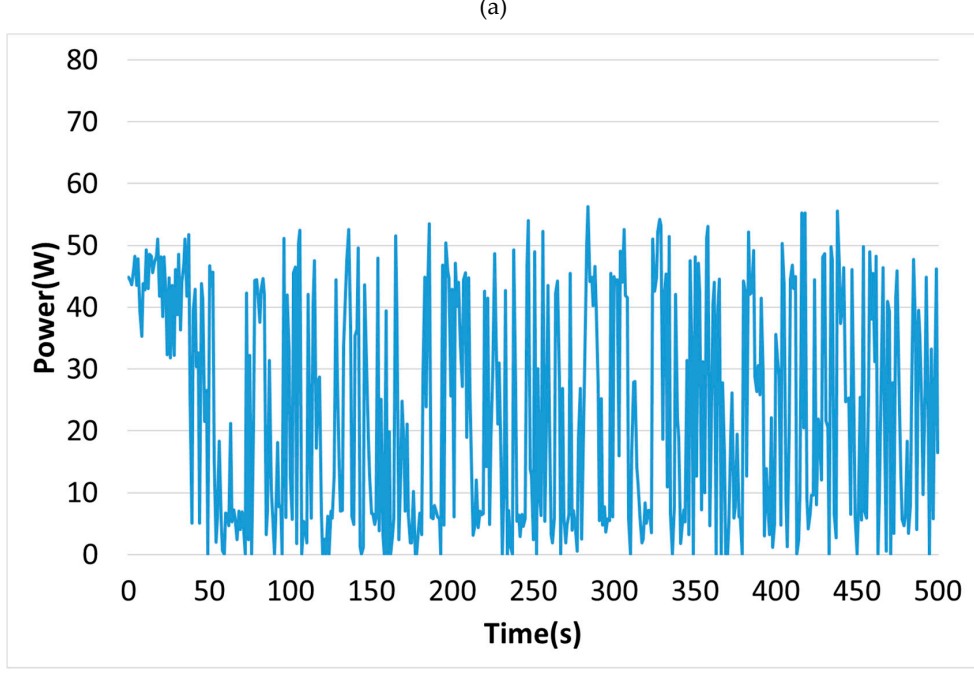

(b)

**Figure 13.** Voltage, Current, and Power of SF-PI controller. (**a**) Voltage and Current. (**b**) Power.

**Table 7.** Comparison of average power consumption.

|  | PI Controller | SF-PI Controller |
|---|---|---|
| Average Power Consumption (Wh) | 25.24 | 23.45 |

The temperature control of the thermoelectric element controls the cooling of the thermoelectric element by the supply of electric power, and when the temperature of the heat absorption side in the thermoelectric element is lower than the set value, that is controlled to rise by the supplied air.

Figure 14 shows the temperatures of the thermoelectric elements of PI and SF-PI when the temperature of the surface of the thermoelectric element reaches a steady state. Figure 14a shows the temperature of the PI controller and Figure 14b shows the temperature of the SF-PI controller. According to the SF-PI operation, the actual value can be divided into the section lower than the set value and the section where the actual value is larger than the set value. If the actual value is lower than the set value, control is required so that the actual value follows the set value quickly. To increase the set value, it increases the error input to the PI controller. Since the error value increases, the control value output from the PI controller rises to enable quick control. If the actual value is higher, control is performed in the opposite direction.

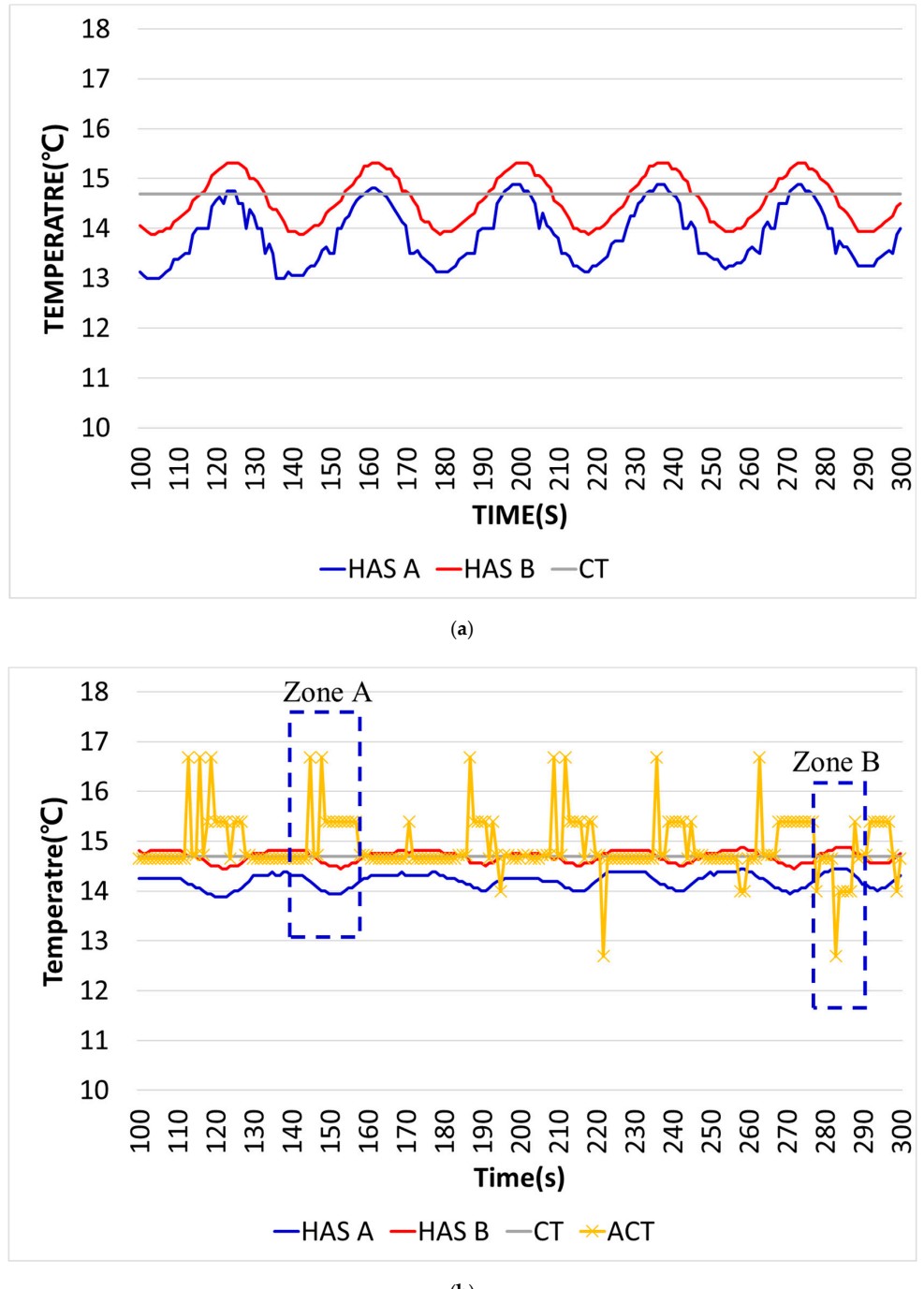

(a)

(b)

**Figure 14.** Temperature comparison in steady-state. (**a**) PI Controller. (**b**) SF-PI Controller.

In Figure 14b, Zone A shows a lower temperature of the heat absorption side (HAS) of the thermoelectric element than the control temperature (CT) and Zone B shows a higher HAS temperature than CT. In Zone A, because it requires a rise in the HAS temperature, the temperature adjusted by the fuzzy control (ACT) rises. In Zone B, the ACT is going down for decreasing HAS temperature.

Figure 15 shows the temperature error between the Control Temperature (CT) and the thermoelectric element temperature. The conventional PI controller uses fixed gains and an input value. However, the SF-PI controller proposed in this paper uses PI's input value adjusted by the fuzzy control according to a control state. Therefore, the SF-PI controller has a lower temperature error than the PI controller. Table 8 shows the average temperature error between the control temperature and the thermoelectric element temperature. In Figure 15, the SF-PI controller performs more accurate temperature control than the PI controller. For this reason, the average temperature error of SF-PI controller is 22% of PI's value.

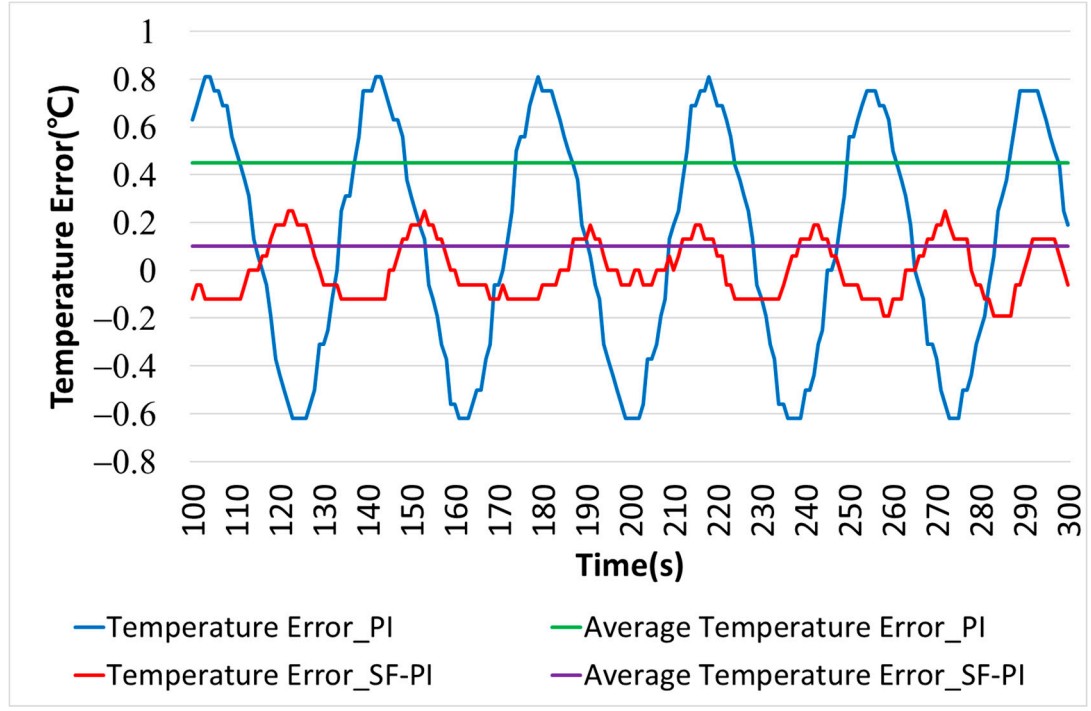

**Figure 15.** Temperature error comparison.

**Table 8.** Temperature error comparison.

|  | PI Controller | SF-PI Controller |
| --- | --- | --- |
| Average Temperature Error (°C) | 0.45 | 0.1 |

Figure 16 shows the consumption power and the average consumption power of the PI controller and the SF-PI controller under the conditions of Table 4. The consumption power of thermoelectricity is generated in a cooling period. Therefore, to decrease its consumption power, the cooling time has to be reduced. The SF-PI controller has a shorter cooling time than PI because the temperature error is low. Therefore, the SF-PI controller has a short cooling time, and the consumption power is reduced.

Table 9 shows the average consumption power in Figure 16.

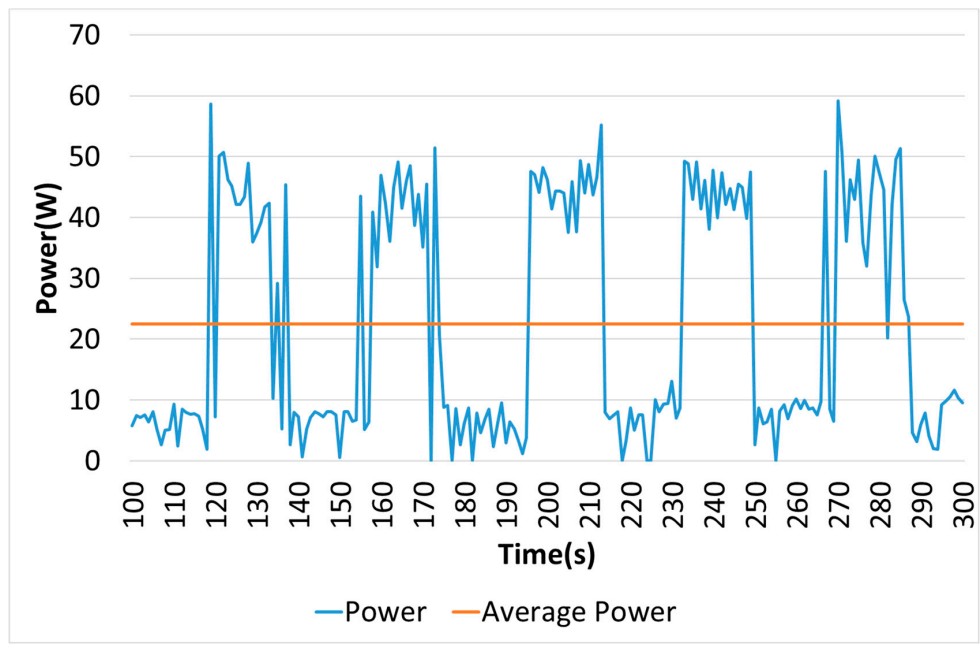

(**a**)

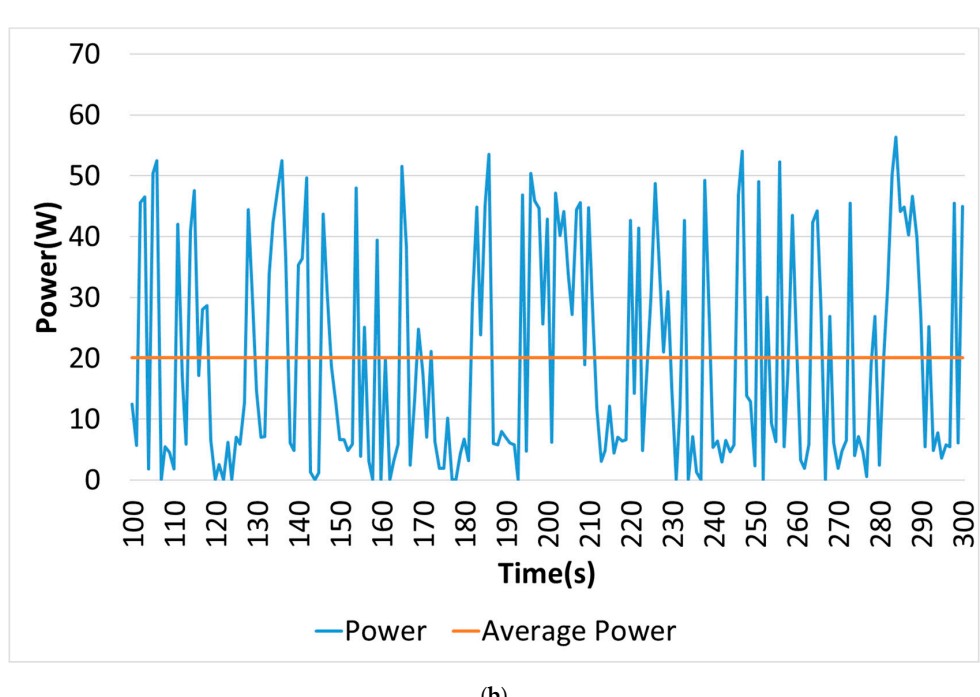

(**b**)

**Figure 16.** Consumption power comparison in steady-state. (**a**) PI Controller. (**b**) SF-PI Controller.

**Table 9.** Average consumption power comparison in steady-state.

|  | **PI Controller** | **SF-PI Controller** |
|---|---|---|
| Average Power Consumption (Wh) | 22.5 | 20.1 |

The condition of Table 6 is a constant indoor environment. To analyze changing environments, Figure 17 shows an indoor environment condition where the dew point temperature is increasing according to decreasing relative humidity caused by indoor temperature increasing.

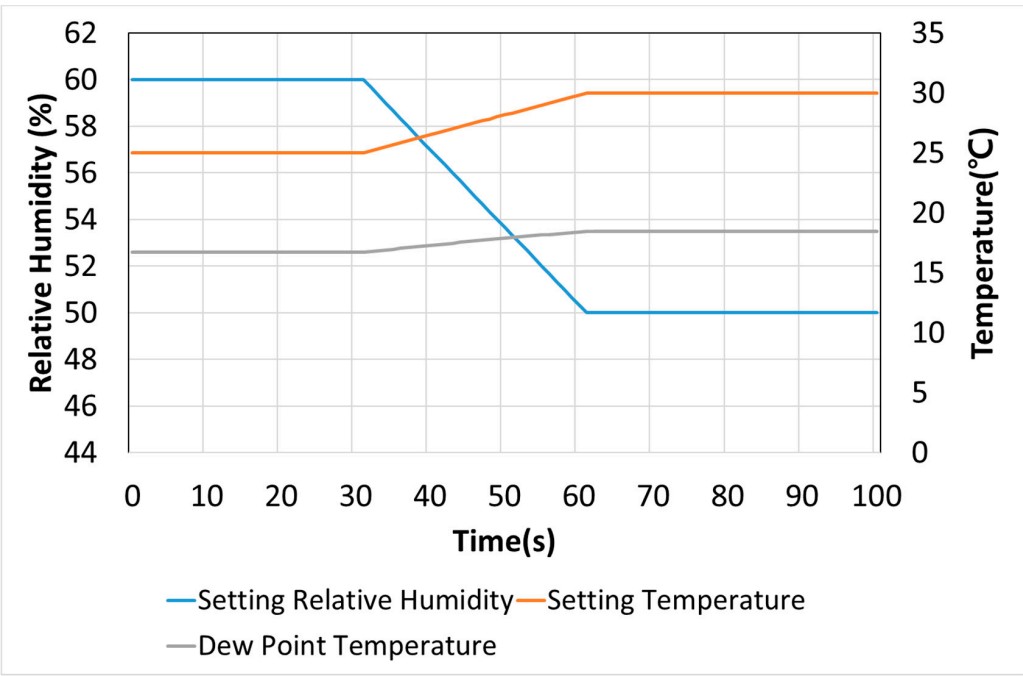

**Figure 17.** Condition of changing temperature and humidity.

Figure 18 shows a response characteristic of PI and SF-PI controller at Figure 17 conditions. The control temperature (CT) is changed by the dew point temperature changing and the PI and SF-PI controller performs temperature control for tracking the CT.

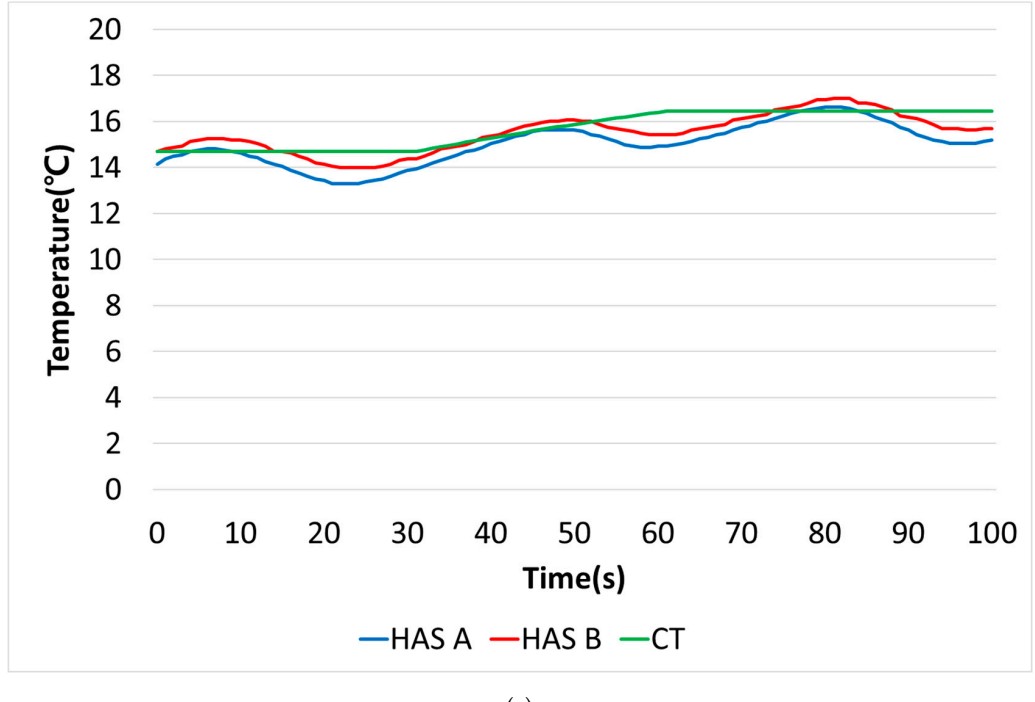

(**a**)

**Figure 18.** *Cont.*

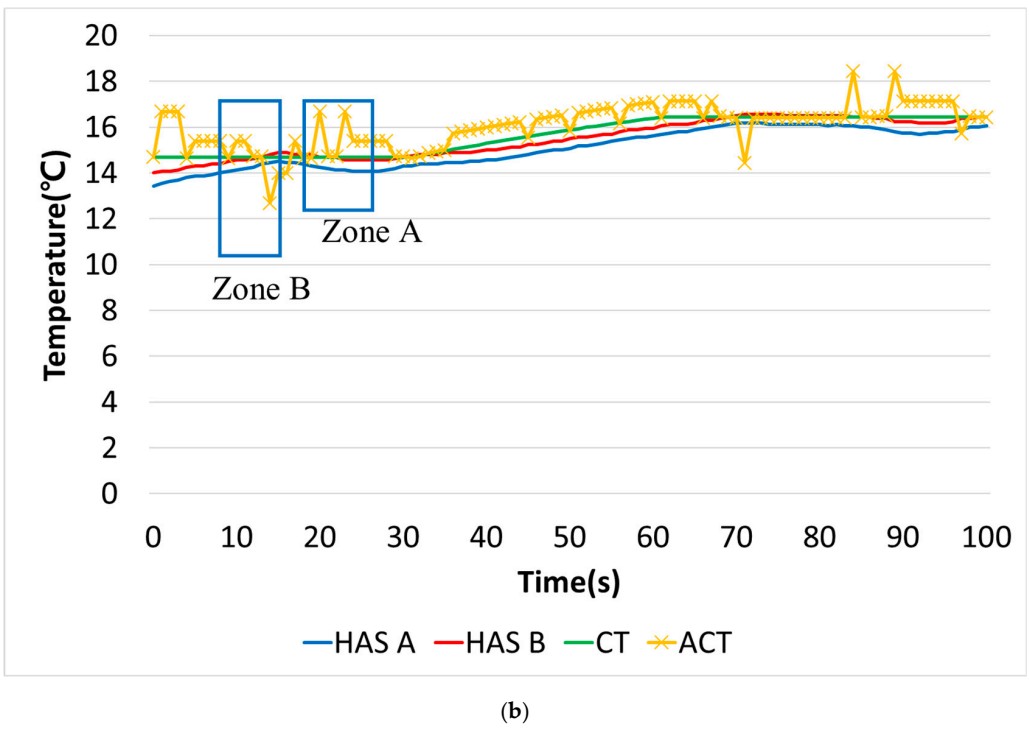

(**b**)

**Figure 18.** Response characteristics by condition of changing temperature and humidity. (**a**) PI Controller. (**b**) SF-PI Controller.

In Figure 18b, Zone A is an area where a heat absorption side (HAS) temperate of thermoelectricity is lower than the CT. In this case, to reduce unnecessary cooling time, adjusted control temperature (ACT) increases, and because of this, the power feeding to thermoelectricity can stop quickly.

Zone B in Figure 18b is a region wherein the HAS temperature is higher than the CT. In this condition, the performance of dehumidification weakens because of insufficient cooling. Therefore, to improve the dehumidification performance, the HAS has to cool quickly. To meet this requirement, the ATC is decreasing, can obtain fast cooling of HAS.

Figure 19 shows a temperature error of the PI and the SF-PI controller under Figure 17 condition. The SF-PI controller can also achieve accurate temperature control under conditions of changing temperature and relative humidity by adjusting PI's input value by fuzzy control. Table 10 shows the result of Figure 19. The temperature error of the SF-PI controller is reduced by about 50% compared to the PI controller.

Figure 20 shows the consumption power of SF-PI and PI controller under condition of Figure 17. Because the SF-PI controller has a low temperature error, its consumption power is also low under conditions of changing temperature and relative humidity. Table 11 shows analysis of Figure 20. The SF-PI controller reduces consumption power by about 2.3% compared to PI controller even in condition of Figure 17.

In the power consumption comparison of Tables 7, 9, and 11, the difference between the PI controller and the SF-PI controller is 1.79 Wh, 2.4 Wh, and 0.5 Wh, respectively. The difference between these values seems to be very small from a typical numerical comparison. However, this is due to the small capacity of the system, and a percentage comparison shows a 7.6%, 11.9%, and 2.3% reduction in power consumption, respectively. Therefore, if the system capacity is increased or the used number increases and the usage time and duration are increased, the effect will be increased.

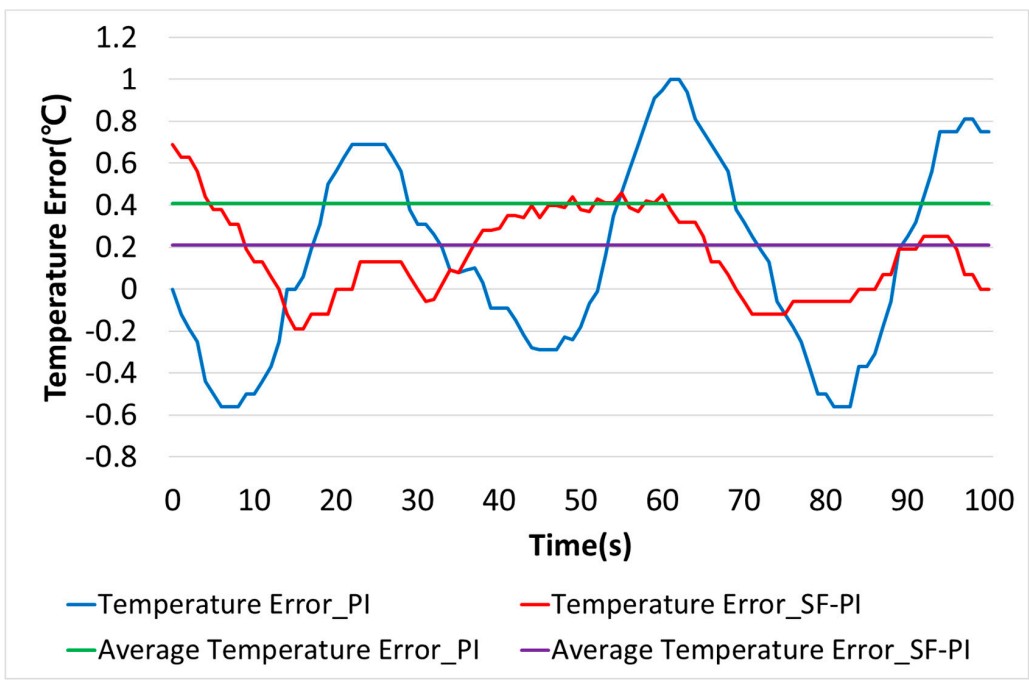

**Figure 19.** Temperature error comparison for changing temperature and humidity.

**Table 10.** Comparison of average temperature error about changing temperature and humidity.

|  | PI Controller | SF-PI Controller |
|---|---|---|
| Average Temperature Error (°C) | 0.41 | 0.20 |

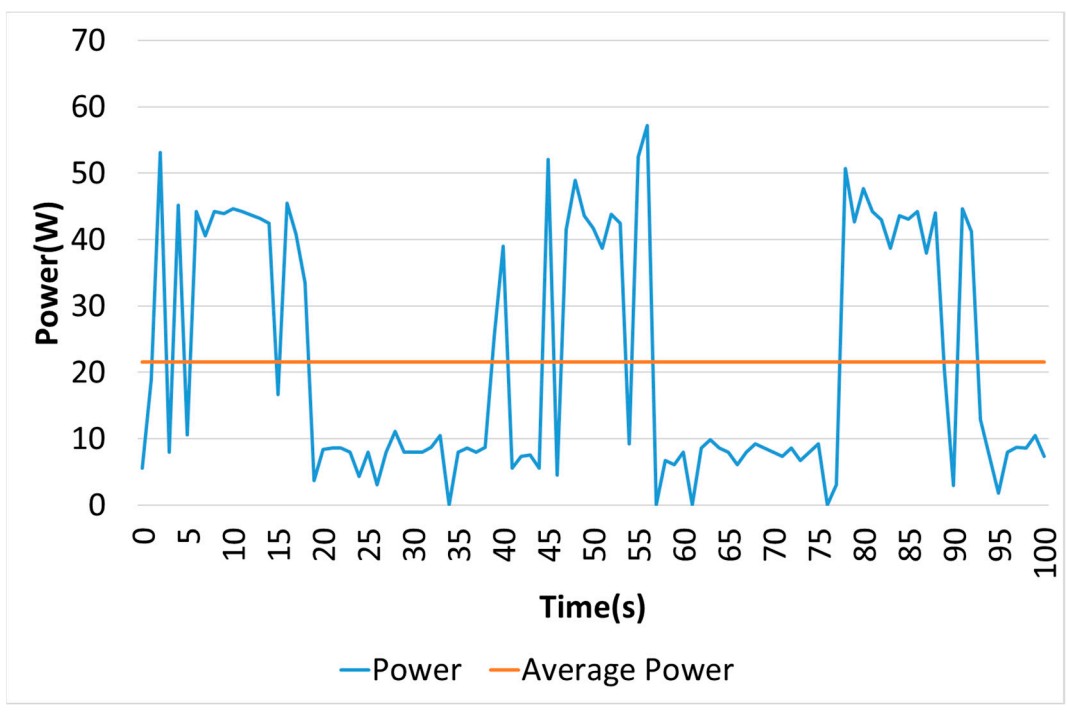

(**a**)

**Figure 20.** *Cont.*

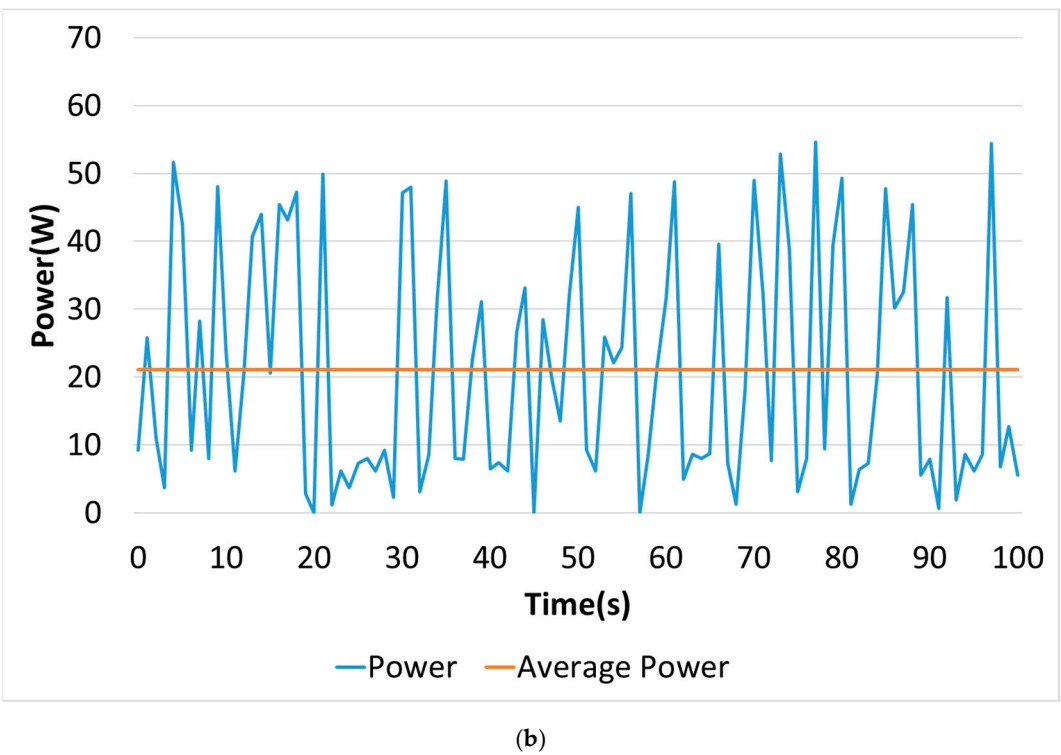

(**b**)

**Figure 20.** Consumption power comparison for changing temperature and humidity. (**a**) PI Controller. (**b**) SF-PI Controller.

**Table 11.** Average consumption power comparison for changing temperature and humidity.

|  | PI Controller | SF-PI Controller |
|---|---|---|
| Average Power Consumption (Wh) | 21.5 | 21.0 |

## 5. Conclusions

This paper proposes applying the IoT to dehumidifier, which is one of the Industry 4.0 techniques, and the method to improve the performance of dehumidifier using the thermoelectric element. To dehumidify using the thermoelectric element, the temperature of the heat absorption side in the thermoelectric element must remain under the dew point temperature. This condition is very closely connected to the dehumidification performance of the dehumidifier.

This paper proposes the SF-PI controller consisting of fuzzy control and PI controller connection. The conventional PI controller is not easily satisfied in both transient-state and steady-state because it is controlled using a fixed gain. However, the SF-PI controller can be expected to achieve satisfactory control performance compared to the conventional PI control because it controls the temperature using the adjusted value by fuzzy control according to the operating conditions.

The average temperature error of SF-PI controller between the reference temperature and the thermoelectric element temperature is 22% of traditional PI's value. It also decreases the consumption power by about 10% by prevention of overwork cooling. Despite temperature and humidity variation, the temperature error and the consumption power of the SF-PI controller is respectively reduced by about 50% and 2.3% compared to the PI controller.

Therefore, the SF-PI controller proposed in this paper has excellent control performance compared to PI control, and therefore can improve the performance of a dehumidifier using a thermoelectric element. In addition, the IoT, part of Industry 4.0, of the dehumidifier is performed according to remote control by a web page. By applying Industry 4.0 to the dehumidifier system, it is possible to

provide remote control and monitoring function to the user, and it is possible to analyze the usage pattern of the user and provide control operation tailored to the user.

This paper presents the improvement of the performance of the dehumidifier system using thermoelectric devices and the application of Industry 4.0. The most important performance in a dehumidification system is the ability to maintain the temperature for dehumidification. In this paper, we propose a SF-PI controller and confirm that the temperature control performance is improved compared to the conventional PI controller, and the power consumption of the dehumidification system is also reduced. Recently, the use of dehumidifying systems has been increasing because of the growing interest in the indoor environment. The system proposed in this paper has improved the performance and power consumption of the dehumidification system and can be used as a system that can provide convenience to users through the application of Industry 4.0.

This paper improves the temperature control performance of dehumidifier using a thermoelectric element by SF-PI controller. The SF-PI controller performs switching operation more frequently than a conventional PI controller for accurate temperature control. This results in a reduction in power consumption. Therefore, research to improve performance of the power loss and temperature control by optimized switching operating is necessary.

**Author Contributions:** Conceptualization, J.-S.K.; Data curation, J.-S.K., Jun-Ho Huh and J.-C.K.; Formal analysis, J.-S.K.; Funding acquisition, J.-S.K. and J.-H.H.; Investigation, J.-S.K. and J.-H.H.; Methodology, J.-H.H. and J.-C.K.; Project administration, J.-H.H. and J.-C.K.; Resources, J.-H.H. and J.-C.K.; Software, J.-H.H.; Supervision, J.-C.K.; Validation, J.-C.K.; Visualization, J.-C.K.; Writing—original draft, J.-S.K. and J.-H.H.; Writing—review & editing, J.-H.H. and J.-C.K.

**Conflicts of Interest:** The authors declare no conflict of interest.

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
