# Peer review of "Improvement of Temperature Control Performance of Thermoelectric Dehumidifier Used Industry 4.0 by the SF-PI Controller"

_processes, doi:10.3390/pr7020098_

Reviewer 1 Report

The efforts carried out by the authors are appreciable in the revised manuscript. Most of the provided suggestions have been addressed in a proper way. Therefore, after a careful revision, this reviewer considers that the revised manuscript has been noticeably enhanced and is suitable for publication in Processes in present form.

Author Response

Reply

-Thank you for your appropriate comment. The contribution of this research work has been newly added to the body, highlighting it in red. Thus, I respectfully would like to request your re-review if possible.

Reviewer 2 Report

The work has been improved. However, it still has deficiencies and flaws:

- The English still needs to be improved. Some examples (there are much more):

79-  "for reduce". It should be for reducing
83 -"Therefore, the methods need researching which made reduces dependence for the gain value of PI controller". Made reduces?!
133 "range stability performance"?
135-"However, the  excessive  values of these  bring the  feedback  value  of system
continual increasing or decreasing and caused the vibration of system".  This sentence is not understood. What is feedback value?

- There are no captions for tables in lines 139 and 141. The first table summarizes the general effects of the PI parameters on minimal phase processes. However, there is no knowlegde of the behavior of the system under study, so the effects can be different.
The second table does not  provide interesting information since the methods in the table are not used in the work. The tuning procedure is manual by trial and error.

- The proposed control scheme of figure 4 is still not clear.

- figures 12b and 13b (power) are the same.

- According to tables 5,7 and 9, the power consumption of PI controller and SF-PI controller is very similar. Therefore, the improvement of the more complex control methodology is not well justified.

- What is the advantage of using Industry 4.0 instead of a conventional control system implementation? The work should be focused on either the advantages of the proposed SF-PI control, or the advantages of using indsutry 4.0 technology.

Author Response

Reply

-First of all, we really appreciate your re-review. Following your comment, the contribution of the study has been added to paper while making modifications with the help of a native English speaker. Thus, we’d be most grateful if you will review our research work again. The additions and changes made are being highlighted in red and blue.

Yes Can be improved Must be improved Not applicable
Does the introduction provide sufficient background and include all relevant references? (x) ( ) () ( )
Is the research design appropriate? ( ) (x) ( ) ( )

Reply

- In order to improve the clarity of the system presented in this paper, we have added a description of Figure. 4

ADD 1)

The purpose of this paper is to improve the temperature control performance of a dehumidifying system using a thermoelectric device. In this paper, we design a system to control the input value of the PI controller because the PI controller has a limitation in the performance improvement due to the fixed gain value. Control of input values of PI controller uses fuzzy control that does not require mathematical modeling and robustness to nonlinear system.

 The PI controller controls the PWM for the thermoelectric device control, and the fuzzy control controls the input value of the PI controller. In this paper, fuzzy control and PI controller are arranged and connected in series to construct a system that operates like this.

The fuzzy control is performed by inputting the error and the error value by the reference temperature (T *) and the heat absorption side temperature of the thermoelectric element. If the dew point temperature (TDP) is used as the reference temperature for the thermoelectric device HAS temperature control, the change in temperature and relative humidity due to the ambient environment will greatly affect the dehumidification performance. Therefore, the reference temperature is set using the band gap temperature (ΔT), and the HAS temperature is controlled below that temperature.

If two or more thermoelectric elements are used in the dehumidifier, the temperature of each HAS is different. The HAS temperature is most closely related to the dehumidification performance, and if the HAS temperature is higher than the dew point temperature, dehumidification is not performed.

Therefore, each HAS temperature is measured and the temperature control is performed based on the higher HAS temperature. A block diagram of this system is shown in Figure 4.

Are the methods adequately described? ( ) ( ) (x) ( )

Reply

-Additional effort was made to describe the methodology adequately.    

Are the results clearly presented? ( ) ( ) (x) ( )

Reply

- We replaced the duplicate picture with the result and added a description of the figure.

CHANGE 1)

(b) Power

Figure 13. Voltage, Current and Power of SF-PI controller

 ADD 1)

In the steady state, the decrease of the power consumption is shown by the SF-PI controller being 2.4Wh lower than the conventional PI controller. The average power consumption of PI controller and SF-PI controller is 22.5Wh and 20.1Wh respectively, 2.4Wh shows about 10% improvement. If the system becomes larger, and the time and duration of use become longer, the difference will be even greater.

Are the conclusions supported by the results? ( ) ( ) (x) ( )

Reply

-I’ve revised the contents to support Conclusion based on the results obtained. The added content is as in the following.

ADD 1)

By applying Industry 4.0 to the dehumidifier system, it is possible to provide remote control and monitoring function to the user, and it is possible to analyze the usage pattern of the user and provide the control operation tailored to the user.

This paper presents the improvement of the performance of the dehumidifier system using thermoelectric devices and the application of the industry 4.0. The most important performance in a dehumidification system is the ability to maintain the temperature for dehumidification. In this paper, we propose a SF-PI controller and confirm that the temperature control performance is improved compared to the conventional PI controller, and the power consumption of the dehumidification system is also reduced. Recently, the use of dehumidifying system has been increasing because of the growing interest in the indoor environment. The system proposed in this paper has improved the performance and power consumption of the dehumidification system and can be used as a system that can provide convenience to users through application of Industry 4.0.

Comments and Suggestions for Authors

The work has been improved. However, it still has deficiencies and flaws:

- The English still needs to be improved. Some examples (there are much more):

79-  "for reduce". It should be for reducing

83 -"Therefore, the methods need researching which made reduces dependence for the gain value of PI controller". Made reduces?!

133 "range stability performance"?

135-"However, the  excessive  values of these  bring the  feedback  value  of system

continual increasing or decreasing and caused the vibration of system".  This sentence is not understood. What is feedback value?

Reply

-The reviewer's comment was revised and supplemented as follows.

CHANGE 1)

Especially, the PI control has excellent advantage for reducing error in steady-state [28].

CHANGE 2)

Therefore, it is necessary to study the method for reducing the dependence of gain of PI controller[29].

CHANGE 3)

The kp and ki (Ti) must be adjusted within a stability performance range.

CHANGE 4)

However, excessive increase or decrease of these values causes vibration of the system.

- There are no captions for tables in lines 139 and 141. The first table summarizes the general effects of the PI parameters on minimal phase processes. However, there is no knowledge of the behavior of the system under study, so the effects can be different.

The second table does not provide interesting information since the methods in the table are not used in the work. The tuning procedure is manual by trial and error.

Reply

-The reason for adding the general effect of the gain value of the PI controller is to explain the effect of the gain of the PI controller on the system and to explain how to set the gain of the PI controller in this paper.

In the second table (Table 2), the trial and error method used in this paper is the manual method among the various methods for controlling the gain value of the PI controller. This is to show the difference between this method and the other methods.

ADD 1)

Table 2 shows the various ways to adjust the gain of the PI controller [33-34]. This paper adjusts the gain of the PI controller through the trial and error method, which is one of the manual methods that do not need mathematical expression and can control the gain value online.

- The proposed control scheme of figure 4 is still not clear.

Reply

-To clarify the structure of the proposed SF-PI controller, we have added the following information.

ADD 1)

The purpose of this paper is to improve the temperature control performance of a dehumidifying system using a thermoelectric device. In this paper, we design a system to control the input value of the PI controller because the PI controller has a limitation in the performance improvement due to the fixed gain value. Control of input values of PI controller uses fuzzy control that does not require mathematical modeling and robustness to nonlinear system.

 The PI controller controls the PWM for the thermoelectric device control, and the fuzzy control controls the input value of the PI controller. In this paper, fuzzy control and PI controller are arranged and connected in series to construct a system that operates like this.

- figures 12b and 13b (power) are the same.

Reply

-We have replaced the duplicated Figure 13 (b) with another Figure.

CHANGE 1)

(b) Power

Figure 13. Voltage, Current and Power of SF-PI controller

 - According to tables 5,7 and 9, the power consumption of PI controller and SF-PI controller is very similar. Therefore, the improvement of the more complex control methodology is not well justified.

Reply

-The capacity of the system used in this paper is very small. Thus, the difference in power consumption between the PI and SF-PI controllers may be numerically lower. However, if the difference is expressed as a percentage, the improvement effects of Figure 7, 9, and 11 are 7.6%, 11.9%, and 2.3%, respectively. We have added these contents to the paper as follows.

ADD 1)

 In the power consumption comparison of Tables 7, 9 and 11, the difference between the PI controller and the SF-PI controller is 1.79 Wh, 2.4 Wh and 0.5 Wh, respectively. This shows 7.6%, 11.9%, and 2.3% reduction in system power. Simple power consumption is low because of the small system capacity, but if the capacity of the system is increased, and the time and duration of use are increased, the effect will increase.

- What is the advantage of using Industry 4.0 instead of a conventional control system implementation? The work should be focused on either the advantages of the proposed SF-PI control, or the advantages of using indsutry 4.0 technology.

Reply

-The benefits of using Industry 4.0 and the focus of the paper have been added to the conclusion section as follows:

ADD 1)

By applying Industry 4.0 to the dehumidifier system, it is possible to provide remote control and monitoring function to the user, and it is possible to analyze the usage pattern of the user and provide the control operation tailored to the user.

This paper presents the improvement of the performance of the dehumidifier system using thermoelectric devices and the application of the industry 4.0. The most important performance in a dehumidification system is the ability to maintain the temperature for dehumidification. In this paper, we propose a SF-PI controller and confirm that the temperature control performance is improved compared to the conventional PI controller, and the power consumption of the dehumidification system is also reduced. Recently, the use of dehumidifying system has been increasing because of the growing interest in the indoor environment. The system proposed in this paper has improved the performance and power consumption of the dehumidification system and can be used as a system that can provide convenience to users through application of Industry 4.0.

Round  2

Reviewer 2 Report

The work has been improved

This manuscript is a resubmission of an earlier submission. The following is a list of the peer review reports and author responses from that submission.

Round  1

Reviewer 1 Report

The manuscript proposes a controlled dehumidifier using open source devices like the microcontroller Arduino. A control method combining fuzzy logic and proportional-integral control is applied. The paper requires great efforts to improve its quality and presentation for the prestigious journal Processes. A set of comments are expounded hereafter.

- The manuscript is well written and organized. However, there are some minor mistakes or improvements to make regarding the format of the document, as commented below.

Between the last two authors’ names, “and” must be inserted.

In the affiliation of the authors, their faculty position is not necessary.

In line 30, CO2 appears in cursive style, but there is no reason for that from this reviewer’s perspective.

The English language must be improved. For instance, in line 52 “A methods…” is incorrect.

The point after the word figure is unnecessary.

Table 1 is not compliant with the template of the Journal.

In Figure 2, the units indicated in the vertical and horizontal axes would be better placed between parentheses.

The symbol “&” should be removed in the caption of figure 14a).

In the title of table 3, this reviewer suggests replacing “parts” by “components”.

Concerning the references, they must be thoroughly revised according to the template.

- About the content of the manuscript, the comments after a careful revision are the following:

In the keywords, an interesting term to be included is “temperature control”, if the authors agree with the suggestion. Indeed, “open source” could also be added.

In the first section, the contextualization of the proposal must be enhanced. Namely, the Industry 4.0 should be more commented. It comprises a number of advanced functions and has several implications. Given the fact that the title includes explicitly the term “Industry 4.0 technique”, the authors should clearly explain what that technique is.

Aligned with the previous, the authors should comment about the concept of “smart home”, which is involved within the Industry 4.0 paradigm. Moreover, other related concepts are “smart buildings”. The background of the manuscript has to be improved in order to highlight the advanced scenarios where the proposal is framed.

Indeed, intelligent control methods, like fuzzy control, play a paramount role in the Industry 4.0 arena; therefore, the authors should also mention this issue for a better understanding of their contribution.

In a similar sense, the role of open source in the Industry 4.0 should be briefly introduced given the fact that open source tools are used in the proposal of the manuscript. The following recent publications can be considered by the authors:

-          Martinez, B.; Vilajosana, X.; Kim, I.H.; Zhou, J.; Tuset-Peiró, P.; Xhafa, A.; Poissonnier, D.; Lu, X. I3Mote: An Open Development Platform for the Intelligent Industrial Internet. Sensors 2017, 17, 986. DOI: 10.3390/s17050986.

-          Syafrudin, M.; Fitriyani, N.L.; Li, D.; Alfian, G.; Rhee, J.; Kang, Y.-S. An Open Source-Based Real-Time Data Processing Architecture Framework for Manufacturing Sustainability. Sustainability 2017, 9, 2139. DOI: 10.3390/su9112139.

-          Viciana, E.; Alcayde, A.; Montoya, F.G.; Baños, R.; Arrabal-Campos, F.M.; Zapata-Sierra, A.; Manzano-Agugliaro, F. OpenZmeter: An Efficient Low-Cost Energy Smart Meter and Power Quality Analyzer. Sustainability 2018, 10, 4038.

What does this sentence mean “Also, the Industry 4.0 verifies by remote control and a monitoring in real time”? It is confusing and should be rewritten. The same issue occurs with this sentence of the Conclusions “…the Industry 4.0 of dehumidifier performed according to control remote by web page.”

Moreover, a common practice in scientific papers is to include a brief paragraph at the end of the Introduction in order to indicate the structure of the document. This helps the reader to have an accurate idea about the organization and facilitates the reading. This reviewer suggests including such a paragraph to enhance the manuscript presentation.

In lines 133-134, the labels of the fuzzy subsets are decomposed. However, from this reviewer’s viewpoint, this explanation should be given as text, not as isolated information.

Why have the authors selected triangular membership functions? Have they tested other shapes, for instance, trapezoidal?

A positive aspect is the explicit mention of the defuzzification method (center of gravity).

In section 4, the first sentence, “To apply the Industry 4.0, this paper used the Raspberry Pi 3”, also causes certain misleading to this reviewer. Do the authors consider that using a Raspberry Pi fulfills Industry 4.0 requisites? This reviewer agrees about the use of open source technology in the Industry 4.0 facilities. However, the reader should find this consideration in the manuscript.

The initial text of the fourth section should be rewritten in order to clearly describe the components (hardware and software) of the proposed system, as well as their interconnection. In addition, a block diagram or scheme would help to understand such proposed system.

Also, the open source nature of the involved devices (Raspberry Pi and Arduino) and software (MySQL) should be highlighted in the text.

The web page to enable remote control is a very interesting functionality of the developed system.

In the fourth section, the explanations about the changes in the operating state of the dehumidifier do not require so many figures.

The Raspberry Pi seems to play a paramount role in the proposal but it is not included in the reported results, at least, not in an explicit manner. All the attention is paid to the Arduino board.

What about the software involved in the proposal? Apart from the MySQL mention, any other software is mentioned. The Integrated Development Environment (IDE) of Arduino should be mentioned, as well as the libraries that have been used.

The whole section 4 is too long. At least, it should be divided into various subsections.

The provided graphs are scarcely commented whereas there is a large amount of them. This reviewer suggests reducing the amount of graphs and explaining the measured magnitudes with more detail.

In the Conclusions, the reduction of the temperature error is said to be 78%; however, in the fourth section, there is not mention to such value.

As a conclusion of the revision, in its current state, the manuscript must address the provided suggestions to reach a better presentation and scientific level, according to the prestigious journal Processes.

Reviewer 2 Report

The paper proposes the application of a called SF-PI control scheme to a temperature control system. In opinion of this reviewer, the work has a lot of deficiencies and flaws:

- The English is very poor and needs to be deeply revised. There are many typos and grammar errors.

- The references about control must be improved.

- What is the reference for the statement "PID is used about 80% in the industrial field"?

- The work claims that the proposed method is performed using the industry 4.0. However, the application only involves a control loop implemented using a raspberry and Arduino that are communicated via web. This is not necessarily industry 4.0

- There is a lack of control foundation. The PI controller has integral action and therefore the error must be zero in stationary state, assuming closed loop stability. It is said that increasing the kp gain increases the stationary error! It is the opposite in case of stability. There is not a formal analysis of the closed loop stability of the proposed method. In addition, there is no dynamic model of the system to be controlled, the different variables: controlled variable, manipulated variable, disturbance, references are not well defined. The proposed SF-PI control is not well justified.

- The experiments are not well explained. It is not specified how the parameters of the conventional PI controller are tuned. The same for the PI of the proposed SF-PI control.

- The figures of the experiments can be reduced and the figures from 7 to 12 are not interesting and can be removed and substituted by a more general scheme of the system.